# Fusion dynamics of cubosome nanocarriers with model cell membranes

Brendan P. Dyett [1,2], Haitao Yu [1,2], Jamie Strachan[1], Calum J. Drummond[1]* & Charlotte E. Conn[1]*

Drug delivery with nanocarriers relies on the interaction of individual nanocarriers with the cell surface. For lipid-based NCs, this interaction uniquely involves a process of membrane fusion between the lipid bilayer that makes up the NC and the cell membrane. Cubosomes have emerged as promising fusogenic NCs, however their individual interactions had not yet been directly observed due to difficulties in achieving adequate resolution or disentangling multiple interactions with common characterization techniques. Moreover, many studies on these interactions have been performed under static conditions which may not mimic the actual transport of NCs. Herein we have observed fusion of lipid cubosome NCs with lipid bilayers under flow. Total internal reflection microscopy has allowed visualisation of the fusion event which was sensitive to the lipid compositions and rationalized by lipid diffusion. The fusion event in supported lipid bilayers has been compared with those in cells, revealing a distinct similarity in kinetics.

---

[1] School of Science, College of Science, Engineering and Health, RMIT University, Melbourne, Victoria, Australia. [2] These authors contributed equally: Brendan P. Dyett, Haitao Yu. *email: calum.drummond@rmit.edu.au; charlotte.conn@rmit.edu.au

Targeted drug delivery using nanotechnology requires the intercalation of a therapeutic compound within a nano-carrier (NC). NCs act to improve solubility, protect the therapeutic, alter release profiles and circulation times, or promote active targeting. Navigating the complexity of biological environments has meant that extensive research attention has been directed towards the engineering of NCs.[1,2] NC uptake by cells is regulated by a number of endocytosis pathways which envelop and internalise the external NCs.[3] Despite internalisation, transport to lysosomes often leads to NC/therapeutic decomposition and low treatment efficiencies. The delivery of siRNA, DNA and proteins which are prone to denaturing are particularly hampered by these conditions.[4] A separate mechanism to internalisation is membrane fusion. Upon contact, two discrete lipid bilayer motifs fuse, bridging phospholipid domains and mixing internal components. Research on viruses has identified fusion may be induced by membrane proteins.[5] Conversely, simple alterations in lipid profiles have been shown to impart fusogenic properties, allowing liposomes to subvert endocytosis pathways in the absence of fusogenic proteins.[4,6]

Cubic lipid phases have been identified as possible intermediates in membrane fusion.[7–9] It is perhaps not surprising then that dispersed cubic-phase nanoparticles (cubosomes) have emerged as promising fusogenic NCs.[10] Whereas unilamellar liposomes consist of a single outer layer of an amphiphilic bilayer, cubosomes exhibit bilayers conformed to infinite periodic minimal surfaces, classically described as the Schwarz, diamond or gyroid surface. The internal curvature of the lipid bilayers, separated by internalised water channels, lends itself to a high internal surface area which can be exploited to aid in the encapsulation and release of additional polar and non-polar compounds, as well as complex biomolecules.[11] To this end, as a vehicle for targeted drug delivery,[12–14] cubosomes have demonstrated capacity to deliver antimicrobials,[15] MRI contrast agents,[16] proteins[17] and provide chemotherapy.[18,19] Moreover, cubosomes have shown the potential to transverse the blood–brain barrier,[20] which unlocks potential in treating various neurological diseases.[21] Despite the increasing utilisation of cubosomes, their uptake mechanisms via endocytosis or membrane fusion are still under investigation.[22–25] In either case, understanding surface interactions of the cubic phase with cell membranes is paramount. Elucidating this process will further enhance the capacity to engineer targeted therapeutics.

The interaction of cubosomes with surfaces has revealed that surface adsorption is dependent on the surface philicity of the substrate and surface philicity and internal structure of the lipid nanoparticle.[26–28] Cubosomes of monoolein (MO) stabilised by F127, and cubosomes of glycerol dioleate and phosphatidylcholine stabilised by P80, have been observed to adsorb and relax into monolayers on hydrophobic surfaces.[29,30] Conversely, the interaction between the nanoparticles and hydrophilic silica surfaces is minimal owing to electrostatic repulsion between the two negatively charged surfaces. These interactions may also be enhanced by increased stabiliser concentrations, whereas charge screening leads to the attachment of nanoparticle aggregates.[30,31] Likewise, cationic substrate functionality dramatically enhances the adsorption rate and adsorbed quantity.[28,30] Subsequent work revealed that a small portion of cubosomes attach intact before the nanoparticle material spreads homogeneously at a rate determined by surface charge.[30] The adhesion of these nanoparticles has shown to be determined by the particle's internal hydrophobicity and stabiliser properties, yielding important design implications for crop protection[27] as well as for biomedical applications.[32]

The interaction with supported lipid bilayers (SLBs) has been explored to investigate the response with more biorelevant interfaces. MO cubosomes were shown to adsorb and exchange lipid material with dioleylphosphatidylcholine (DOPC) bilayers. After saturation of the bilayer with MO cubosomes, the particles were subsequently released.[33] Later work from the same group revealed that the adsorption varies depending on the bilayer surface coverage. At high surface coverage, a steady state, MO saturated bilayer is achieved composed of ~72% cubosome components. At low surface coverages, this process was preceded by the spreading of cubosome components to fill the defects within the bilayer. Simultaneously, the reduction in the lattice spacing of surface-bound cubosome revealed the exchange of DOPC into the internal structure of the cubosome.[34] This outcome was supported by the interaction of bilayer vesicles and cubosomes.[35] Comparable results were demonstrated for MO cubosomes and 1,2-dipalmitoyl-sn-glycero-3-phosphocholine (DPPC) Langmuir monolayers whereby the cubosomes disintegrated at the interface. The rate and capacity for adsorption were dependent on the ordering of the lipid monolayer. Namely, the incorporation was hindered in more organised layers, and enhanced in porous layers.[36] Parallel results were demonstrated for phytantriol (Phy) cubosomes on 1-palmitoyl-2-oleoyl-sn-glycero-3-phosphocholine (POPC) bilayers. The results showed an accumulation/exchange of materials up to 82% cubic phase material in the bilayer.[37] The rate of interaction was shown to be accelerated with the inclusion of palmitoylphosphatidylserine (PPS) within the cubosome.[37]

Previous studies largely utilised neutron reflectivity, quartz crystal microbalance (QCM) and ellipsometry to investigate changes in the bilayer. These techniques dispense critical information, however, are unable to disentangle interactions of individual cubosome particles. Total internal reflection fluorescence (TIRF) which has been utilised to observe the fusion behaviour of individual viral and liposome particles[38] has not yet been used to observe fusion of cubosomes. In parallel, microfluidics has emerged as an indispensable tool for evaluating nano–bio interactions.[39] The microfluidic environments provide a high-throughput platform and capability to mimic in vivo conditions. The capacity to introduce a controlled flow is invaluable to evaluating NC characteristics such as particle size and particle shape, which are known to have profound impacts and whose behaviour may further vary due to local flow conditions.[40]

Herein, the combination of TIRF and microfluidics is utilised to elucidate the interaction of cubosomes with lipid bilayers, which facilitates the visualisation of individual cubosome fusion events. SLBs are formed within a microfluidic chamber before the introduction of cubosomes within a laminar flow. The results reveal various regimes of behaviour, with several distinctions compared with liposomes and viral particles. The interaction is sensitive to cubosome and bilayer composition, as well as the surrounding media. Under certain conditions, the cubosomes demonstrate selective attachment without any tailored peptide interactions. Finally, the attachment is compared with uptake in small intestine and STO fibroblast cell lines. This work mimics the drug delivery and release process and provides a pathway to probe and tailor cubosome fusion.

## Results

**SLB formation by solvent exchange.** The supported lipid bilayers were formed by solvent exchange[41] or solvent assisted bilayer formation.[42] As illustrated in Fig. 1a, this method describes the initial dispersion of desired lipids, such as DOPC into alcohols. The alcohol is gradually displaced by water, driving the aggregation of monomers and micelles at the interface, subsequently forming a bilayer at high water concentrations.[41] The formation method has shown to be favourable in generating lipid bilayers

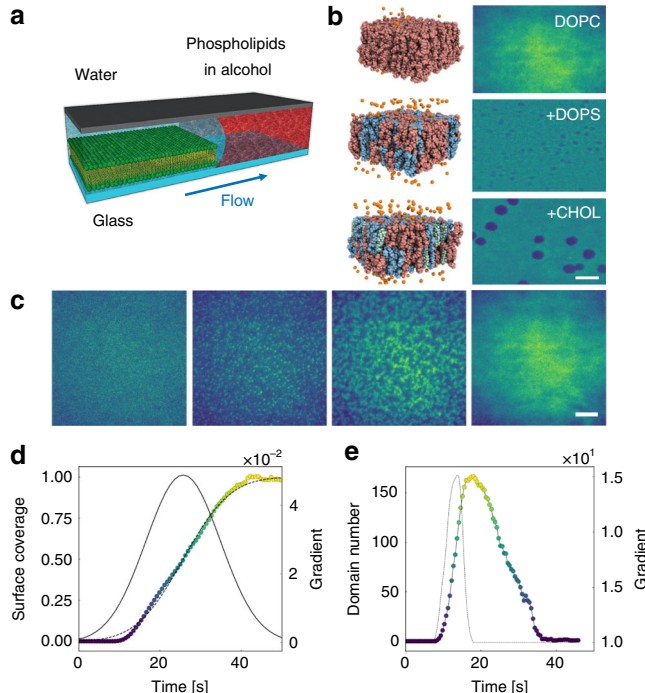

**Fig. 1** Supported lipid bilayer deposition by solvent exchange in a microfluidic channel. **a** Schematic of the solvent exchange process to produce single lipid bilayer on a hydrophilic substrate. **b** Molecular models and the corresponding TIRF results of single lipid bilayers with different components: DOPC, 40:100 DOPS:DOPC and 40:30:100 DOPS:CHOL:DOPC. Scale bar: 10 μm **c** Representative images of lipid bilayer formation on glass substrate, illustrating a nucleation and growth process. Scale bar: 10 μm. **d** Plot of surface coverage of DOPC versus time. The data, error function best fit and best fit gradient are shown as the coloured circles, dashed line and solid line, respectively. **e** Plot of number of DOPC domains versus time and its gradient as a function of time, shown as coloured circles and the dashed line, respectively

within microfluidics and the inclusion of diverse lipid components, including cholesterol (CHOL).[41,42] Example TIRF snapshots of formed bilayers of DOPC, 40:100 DOPS:DOPC and 40:30:100 DOPS:CHOL:DOPC are shown in Fig. 1b. After the formation of a continuous mixed lipid bilayer, distinct domains visible as the darker domains were observed for compositions, including DOPS or CHOL. This was attributed to the contrasting diffusion coefficient of the phospholipid dye within these lipids, and has been observed previously.[42] Additional evidence for the formation of a lipid bilayer was provided by QCM-D and atomic force microscopy (AFM) measurements under identical flow conditions. As shown in Supplementary Fig. 1, QCM demonstrated a frequency shift of ~−25 Hz upon membrane formation, indicating the complete formation of a single lipid bilayer.[42] Moreover, the formation and final state of the lipid membrane was characterised in situ by liquid AFM, shown in Supplementary Fig. 2, showing the same formation process and revealing the expected membrane step height of ~5 nm.

The formation process for a DOPC SLB is shown in Fig. 1 and available in Supplementary Video 1. The initial stages show a direct resemblance to the nucleation of nanodroplets by solvent exchange.[43] Many nanoscopic domains nucleate which continue to grow and fuse together, yielding a continuous SLB. The dynamics of this behaviour is reflected in the plot of surface coverage, and domain number with time shown in Fig. 1d, e, respectively. The former demonstrates a sigmoidal growth

pattern, with an initial lag phase, rapid growth and plateau. The dashed line indicates best fit and gradient error function fit, in good agreement with previous results on droplet volumes.[43] In contrast to the formation of other liquid or solid species by solvent exchange, the domains grow in a fractal manner and consume the available surface area during fusing. Consequently, the domain number with time demonstrates a near gaussian distribution, highlighting a peak early in the exchange process, before reducing to a singular domain.

The nucleation and growth process of DOPC resembles the Frank–van der Merwe (FM) growth mode. Lipid molecules preferentially grow on the hydrophilic glass substrate resulting in smooth, fully formed layers. The layer-by-layer growth is two-dimensional, indicating that complete film forms prior to growth of subsequent layers. It was observed that additional bilayers could form during the solvent exchange, as shown in Supplementary Fig. 3. The fluorescence intensity of the second layer demonstrated a step-wise increase, i.e., intensity of the second layer was double the initial single layer. Over time, the secondary layer showed poor stability and washed away with continued flushing. To ensure formation of a single lipid bilayer, the bilayer was continuously flushed with water for 30 min.

**Cubosome fusion with model cell membranes.** Following the formation of the SLB, a solution of cubosomes was introduced, as illustrated in Fig. 2a, b. The studied cubosomes had internal structure of $Q_{II}P$ phase, as indicated by small-angle X-ray scattering (SAXS) (Supplementary Fig. 4). The cubosomes were loaded with a hydrophobic fluorescent dye, which may be considered analogous to a small hydrophobic therapeutic. Observed as small bright features against the darker SLB background, the cubosomes revealed fascinating and distinct behaviours. These dynamics are best observed in Supplementary Videos 1–3. Time lapses for the cubosome interaction with three bilayer systems are provided in Fig. 2c–e. Within Fig. 2c, MO cubosomes were introduced to a DOPC layer in water. Upon reaching the SLB, the fusion of the cubosome was observed as a bright flash of fluorescence which gradually expanded from the centre. In Fig. 2d, MO cubosomes doped with 1% DOTAP were introduced to a DOPC/DOPS bilayer in water. In this case, the cubosomes attached to either the continuous DOPC rich (bright) or discrete DOPS rich domains (dark spots). While the fusion into the DOPC domains resembles that of just pure DOPC, the fusion into the DOPS rich domains was markedly slower. In contrast to the rapid flashes observed for pure DOPC, interaction of the cubosome with the DOPS-rich domain resulted in a slow increase in nearby fluorescence. Moreover, the cubosomes at the edge of these two phases show an asymmetric release profile (Supplementary Fig. 5). As more cubosomes interacted with the bilayer, the DOPS domains appeared to be consumed, replaced by a continuous singular layer, consisting of DOPS, DOPC and MO. In Fig. 2e, the same bilayer and cubosomes were introduced in PBS buffer. Remarkably, cubosomes only attached to the DOPS rich domains. Cubosomes continually land selectively in these domains, until once again, the domains were consumed yielding a continuous bilayer. The initial delayed adsorption and mixing on DOPS domains can be attributed to relative energetic barriers. The fusion of lipid layers is driven by favourable interactions of the internal hydrophobic contents.[44] Whilst having the highest potency to attract the oppositely charged cubosomes, it is expected these domains resist fusion owing to a hydration barrier. Similarly, it may be expected that packing of these domains is more resilient to pore formation observed by cationic lipids.[45,46] It was observed in Supplementary Figs. 6–7

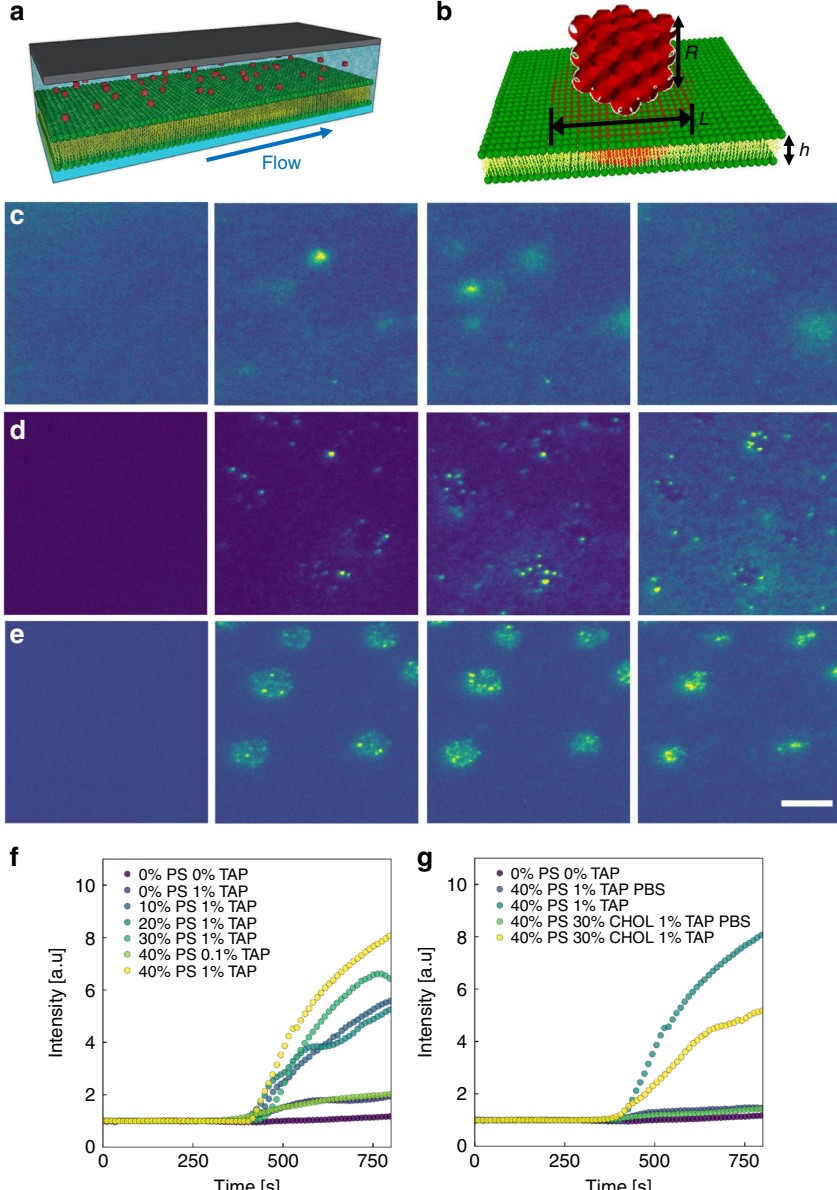

**Fig. 2** Cubosome fusion into lipid bilayer underflow. **a** Illustration of cubosomes in flow above SLB. **b** Illustration of single cubosome above SLB. The interacting system is described by the cubosome length $R$, increasing fluorescent footprint $L$ and bilayer height $h$. **c** TIRF snapshots highlight successive fusion events of MO cubosomes with DOPC bilayer. **d** TIRF snapshots highlight successive fusion events of 1% DOTAP cubosomes with 40% DOPS bilayer. **e** TIRF snapshots highlight successive fusion events of 1% DOTAP cubosomes with 40% DOPS bilayer in PBS. Scale bar = 5 μm. **f** The total fluorescence signal recorded for cubosome introductions with various lipid compositions. **g** The total fluorescence signal recorded for cubosome introductions with various lipid compositions, highlighting impact of PBS environment

that cubosomes could jump to more favourable nearby locations and instantly fuse. In addition, that cubosomes within the DOPS rich domains could persist for long time periods (20 s) before a sudden instability induced a rapid fusion (<1 s).

The lipid packing of DOPC and DOPS bilayers has been shown to vary in the presence of cations. In particular, $Ca^{2+}$ on oxide substrates leads to restricted orientation of serine residues toward the substrate and compression of the bilayer.[47,48] The height step AFM experiments indicate height steps differences ~2 nm between the two domains, as shown in Supplementary Fig. 8. In this case, DOPS-rich regions possibly contain solution-oriented serine residues, inducing the selective attachment

observed. This result highlights that cubosomes may be selective in their adsorption profile, even using only common lipid variants.

Across the time-lapse series, the background fluorescence intensity continually increases as more and more dye is delivered by fusing cubosomes. By analysing the fluorescence intensity with time, the cubosome interactions with the various bilayers were determined. Figure 2f shows the intensity profile for varying bilayer and cubosome compositions. The interactions were tailored by adjusting the negative charge of the bilayer by increasing DOPS moiety (varied from 0 to 40% mol ratio to DOPC). Or increasing the positive charge of the cubosome by

increasing DOTAP moiety (varied from 0 to 1% mol). In all cases, there was an introductory lag, which was the time taken for the cubosomes to reach the substrate in flow. As the magnitude of opposing charges was increased, there was a step-wise increase in the fluorescence profiles, as is reasonably expected. The maximal intensity observed was for the 40% DOPS–1% DOTAP system, which achieved an intensity eight times greater than the standard DOPC—MO system. These intensity profiles indicate that the number and rate of fusion events are enhanced by charge inclusion. On the other hand, when CHOL was included in the bilayer, there was a hampered fluorescence profile, shown in Fig. 1g. Once again, the results within water compared with PBS revealed a stark contrast. While charged lipid motifs showed increased fluorescence, it is markedly reduced compared with their water counterparts. This may be attributed to the increased screening of surface charges reducing the Debye length and the number of collisions.

Whereas the total fluorescence indicated general trends for the system, the high temporal and spatial resolution of TIRF enables elucidation of the behaviour of cubosomes on an individual level. As had been suggested earlier, the cubosomes undergo Brownian motion nearby the surface and in some cases, the cubosomes will stop and release from the substrate.[34] Indeed these various dynamics were visually observed as shown in Supplementary Fig. 7. In comparison, the cubosomes greatly above the substrate, flow uniformly according to the increased velocity away from the channel wall. Within Fig. 3f, the motion of the particles was characterised by their mean-squared displacement (MSD). The linear increase of MSD cubosomes indicates the Brownian motion of the particles across the substrate prior to fusion, captured by the rapid deviation from linearity. The indicative size calculated by the diffusion equation shows reasonable agreement with the size distributions determined by DLS shown in Fig. 3g. The particle size distribution shows a skewed distribution, with the mean size ~200 nm. As the DOTAP composition was increased from 0 to 2%, there was a subtle decrease in particle size, and the zeta potential ($\zeta$) demonstrated an asymptotic increase from near ~0 mV to ~45 mV.

**Fusion dynamics of individual cubosomes with model cell membranes**. A time-lapse of a single cubosome fusion event into a DOPC bilayer is demonstrated in Fig. 3a. A series of individual fusion events under different conditions is included with Supplementary Figs. 7–8. The results for additional phospholipid-based dyes which show the same behaviour are included in Supplementary Figs. 9–11. To capture the dynamics of individual cubosomes, the collected TIRF image stacks were analysed by particle tracking algorithms described within the Experimental section. In each case, the cubosome fusion was characterised according to the peak intensity ($I_{max}$) observed at the centre of the particle. A representative plot of the intensity with time during fusion is shown in Fig. 3b. Upon fixing to the SLB, a peak intensity was observed before decaying over ~6 s. This time taken for $I_{max}$ to reduce to the background intensity will be referred to as the fusion time. The dashed line in Fig. 3b shows the fitting to the function $I = at^{-n}$, which demonstrates strong correlation. Many cubosome fusion events were processed, determining the fitting exponent $n$ and the fusion time. The results for the DOPC – MO system are plotted in Fig. 3c. The fusion time demonstrates a peak ~4.5 s with a skewed Gaussian distribution which correlates to the observed particle size distribution. It was earlier suggested that adsorption is inversely proportional to the diameter of the particle, indicating that the adsorption process is controlled by particle diffusion and includes the diffusion of intact particles towards the interface.[29]

The fitted exponent $n$ shows a distribution around the exponent $n = 0.17$, which is intriguing as it deviates from any typical expected scaling. For standard diffusion, the max concentration, $c_{max}$ should therefore vary with time $t$, as $c_{max} \sim 1/t^{0.5d}$, where d is dimensionality. For 2D diffusion, it would be expected for $n = 1$, i.e., $c_{max} \sim 1/t \sim t^{-1}$, as indicated by the solid red line in Fig. 3c, and as was observed for the diffusion behaviour of fusing vesicles.[49] Here, it is proposed that the reduced exponent was a consequence of the diffusion of the cubosome in its entirety, as opposed to the cubosome acting as a point source release of dye. As depicted in Fig. 2b, the lateral diameter of the increasing fluorescent footprint and bilayer height and the cubosome side length are denoted as $L$, $h$ and $R$, respectively. The fluorescence moiety is considered to be dispersed homogenously within the cubosome, such that the peak intensity $I_{max}$ will scale with $R$. The mass transfer balance during the process of an individual cubosome fusing into single lipid bilayer, which is dominated by diffusion, can be derived by

$$\dot{m} = \rho R^2 \dot{R} = Dh\delta c \qquad (1)$$

where $\rho$ is the density of cubosome, $D$ is the diffusion coefficient of MO in the single lipid bilayer, and $\delta C$ is the concentration gradient of MO in the single lipid bilayer, which is created by an individual cubosome fusion process. According to Fick's law, $\delta c = \frac{c_{max} - c_{L/2}}{L/2}$, where $C_{max}$ and $C_{L/2}$ are the MO concentration at the centre of the cubosome and at the lipid bilayer outside the fluorescent footprint, respectively. As described earlier, for 2D diffusion of an individual particle on a plane, $c_{max} \sim 1/t$. As there is no MO outside the fluorescent footprint, $C_{L/2} \equiv 0$. Then, the lateral diameter of the footprint $L$ scales with $\sqrt{t}$ in the diffusion process, providing a scaling of

$$\delta c = \frac{c_{max} - c_{L/2}}{L/2} \sim \frac{1/t}{\sqrt{t}} = t^{-3/2} \qquad (2)$$

In combination with Eq. (1), a simple ordinary differential equation for $R$ is obtained

$$R^2 \dot{R} = \frac{Dh\delta c}{\rho} \sim \frac{Dht^{-3/2}}{\rho} \qquad (3)$$

which can be integrated over time leading to

$$\int_R^{R_0} R^2 dR \sim \int_0^t \frac{Dht^{-3/2}}{\rho} dt \qquad (4)$$

$$R_0^3 - R^3 \sim -\frac{Dht^{-1/2}}{\rho} \qquad (5)$$

Here, $R_0$ is the initial side length of cubosome, which is a constant during an individual cubosome fusion process. Combining the scaling $I_{max} \sim R$ introduced earlier, a final scaling is obtained,

$$I_{max} \sim R \sim \sqrt[3]{\frac{Dht^{-1/2}}{\rho}} \sim \sqrt[3]{\frac{Dh}{\rho}} t^{-1/6} \qquad (6)$$

indicating an expected scaling of $n = 1/6$, which shows remarkable correlation with the experimental results. This scaling indicates that cubosomes completely fuse with the lipid bilayer instead of acting as a delivery vehicle for dye and shows a contrasting mixing behaviour to that of liposomes.

The fusion of the entirety of the cubosome material was further supported by the clear decrease in fusion time, as DOTAP was included within the formation. Within Fig. 3c, the fusion time shifted to <5 s for 1% DOTAP cubosomes with DOPC, 40% DOPS and 40% DOPS 30% CHOL bilayers. In each case, the peak is only ~0.5–2 s. The acceleration in fusion times is likely a

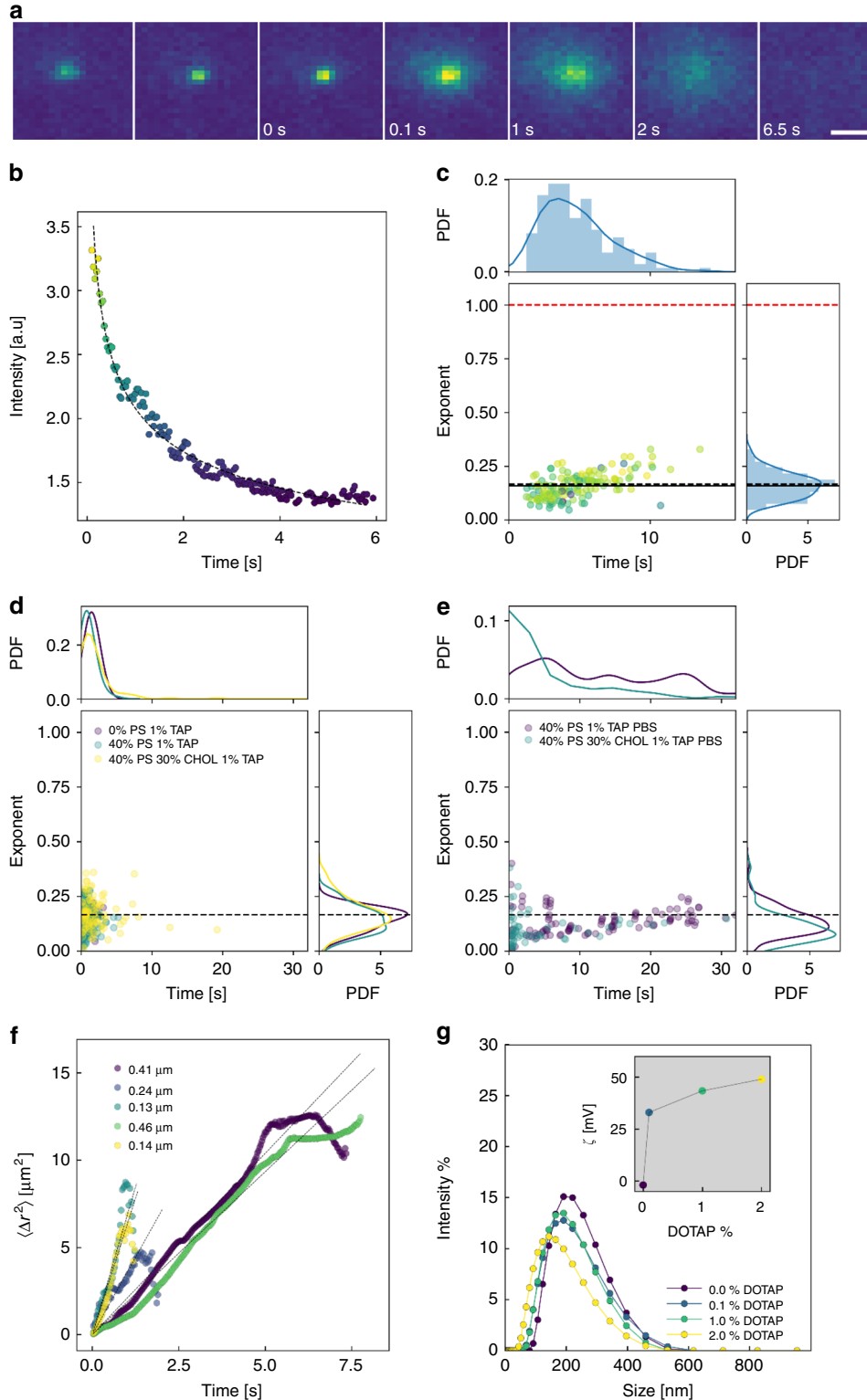

**Fig. 3** Dynamic analysis of individual cubosome fusion. **a** TIRF snapshots of a single cubosome fusion event over the course of ~6 s. Scale bar = 1.5 µm. **b** The corresponding plot of intensity with time during a single fusion event. The best fit to $I = at^{-n}$ is shown by the dashed line; for this case $a$ ~2, $n$ ~0.23. **c** Plot and distribution of fusion time and fitted exponent for many MO cubosome fusion events on a DOPC bilayer. The mean exponent, proposed 1/6 exponent and theoretical 2D diffusion exponent are shown by the solid black, dashed black and solid red lines, respectively. **d** Distribution of fusion time and fitted exponent for 1% DOTAP cubosomes with 0% DOPS, 40% DOPS and 40% DOPS 30% CHOL membranes. **e** Distribution of fusion times and fitted exponents for 1% DOTAP cubosomes with 40% DOPS and 40% DOPS 30% CHOL bilayers in PBS. **f** Measured mean-squared displacement for cubosomes above SLB. The dashed line is best fit against $\langle r^2 \rangle = 2^d Dt$. The determined particle sizes from each slope are shown in the legend. **g** DLS particle size distribution for cubosomes with increasing DOTAP composition. The corresponding zeta potential (ζ) measurements are shown in the inset

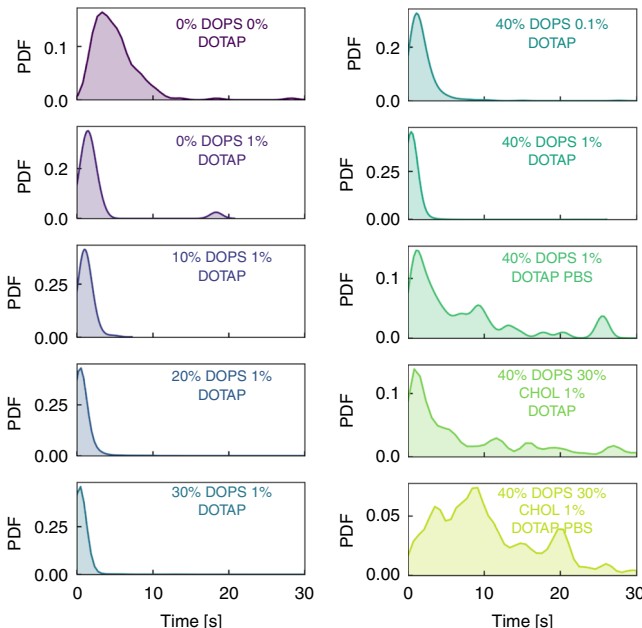

**Fig. 4** Distribution of fusion times measured for varying bilayer and cubosome compositions. In each plot, the distribution is determined by at least 228 unique events

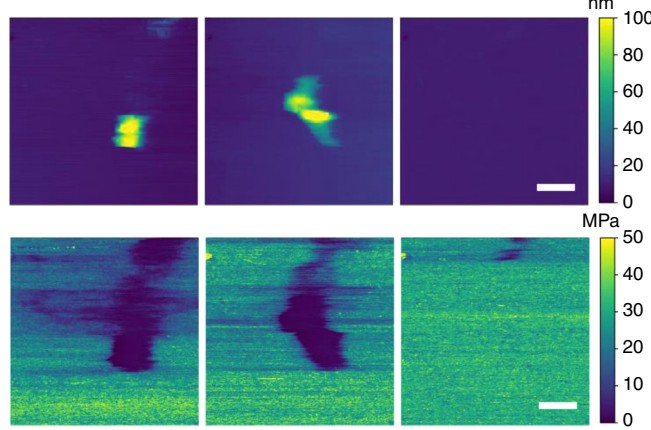

**Fig. 5** Surface measurements during cubosome uptake measured by liquid AFM. AFM sequence showing the variation in surface topology (upper) and Modulus (lower) as a cubosome interacted with SLB. Scale bar = 400 nm

consequence of DOTAP enhancing the fusogenic nature of the cubosome by inducing defects in the bilayer. The reduction in fusion time by an order of magnitude would not be expected in a point source situation. Again, for the charged cubosomes and bilayer, the exponent fitting yields averages within reasonable agreement to the theoretical scaling 1/6. The same analysis performed for PBS samples shows a subtle deviation towards lower exponents, particularly for CHOL containing membranes. This increased deviation is perhaps not unexpected given the multitude of dissimilar fusion profiles and more complex bilayer behaviour.

The fusion times for additional experimental conditions are shown in Fig. 4. For 1% DOTAP cubosomes, as the bilayer composition increased from 0 to 40% DOPS there was gradual reduction, as reflected by the shifting and narrowing distribution peak. The results from CHOL containing bilayers and PBS systems captured the qualitative dynamics described earlier in Fig. 2e. The CHOL bilayer results in a heterogeneous spread of fusion times, dramatically delaying fusion up to ~30 s. Given CHOL should enhance the rigidity of the bilayer, the delayed fusion was somewhat expected. The effect of PBS was even more pronounced. Whereas most interactions finish within 1 s for the 40% DOPS bilayer in water, these times shift to dramatically longer durations within a PBS environment. The distinct adsorption behaviour to the domains shown in Fig. 2e shift fusion times out from several s to a maximum of ~28 s. Once again, this effect is magnified by the presence of CHOL, which demonstrates a higher proportion of longer fusion times and increased heterogeneity across the measured events.

A multitude of complementary analysis was performed on the formed bilayer and cubosome interactions. The cubosome interaction with SLB was probed by liquid AFM. Capturing this process proved exceptionally difficult owing to the short time domains of lipid fusion and the soft mechanical properties for both the particles and SLB. Nonetheless, for charged bilayers in PBS, individual cases of cubosome fusion with the bilayer were observed. An example is shown in Fig. 5. From the height

channel, the ~100 nm feature was observed to spread laterally before disappearing in the final frame. In the modulus channel, the darker region corresponds to the measured modulus of bulk MO cubic phase ~1 MPa[26], whereas the measured SLB modulus ~30 MPa is in line with earlier reports. Most interestingly, the modulus channel demonstrated a significantly larger footprint of low modulus—accompanied by negligible height contrast. This suggests the cubosome had been enveloped within the lipid bilayer and subsequently diffused.

The adsorption of cubosomes into the bilayer was further supported by QCM experiments, which indicated identical trends to those seen in fluorescence studies. In Fig. 6a, the reduction in frequency is reflective of mass increase on the substrate. At the same time, there were sharp increases in the dissipation channel in Fig. 6b. The 3rd, 5th and 7th overtones ($n = 3, 5, 7$) are shown in each plot. Both these results are reflective of MO cubosomes, a very soft material, adhering to the substrate. This is further supported by the overtone dependency for both change in frequency and dissipation. Due to the dependency of the overtone, the Voigt model was used to further examine the viscoelastic properties.[50] Assuming a single layer thin film, the fitting from the three overtones indicated final film thicknesses ranging from 10 to 25 nm, which is in general agreement with earlier reported values.[33] The magnitude of changes determined by QCM were again amplified by incorporating charges to the bilayer and cubosomes. Similarly, their interaction was impeded within buffer environments. Interestingly in Fig. 6c, the plot of dissipation versus frequency revealed a kink for 1% DOTAP cubosomes, which may suggest the rearrangement of lipid material.[51]

Previous results studying these interactions have suggested a mutual exchange in lipid material between adhering cubosomes and the SLB.[31] In this work, there has been a focus on the initial stages of adsorption, which clearly indicate complete fusion of the MO material. By allowing continual cubosome adsorption, the results began to be more reflective of earlier studies. The number of cubosome fusion events was reduced, and attached cubosomes show greatly increased longevity. A measurable exchange in lipid material between SLB and cubosome was observed by introducing separate coloured dyes to each. As shown in Supplementary Fig. 12, the green dye in the SLB accumulated within the MO domain, while red cubosome dye released into the bilayer. At these latter stages, the membrane has become saturated by MO.

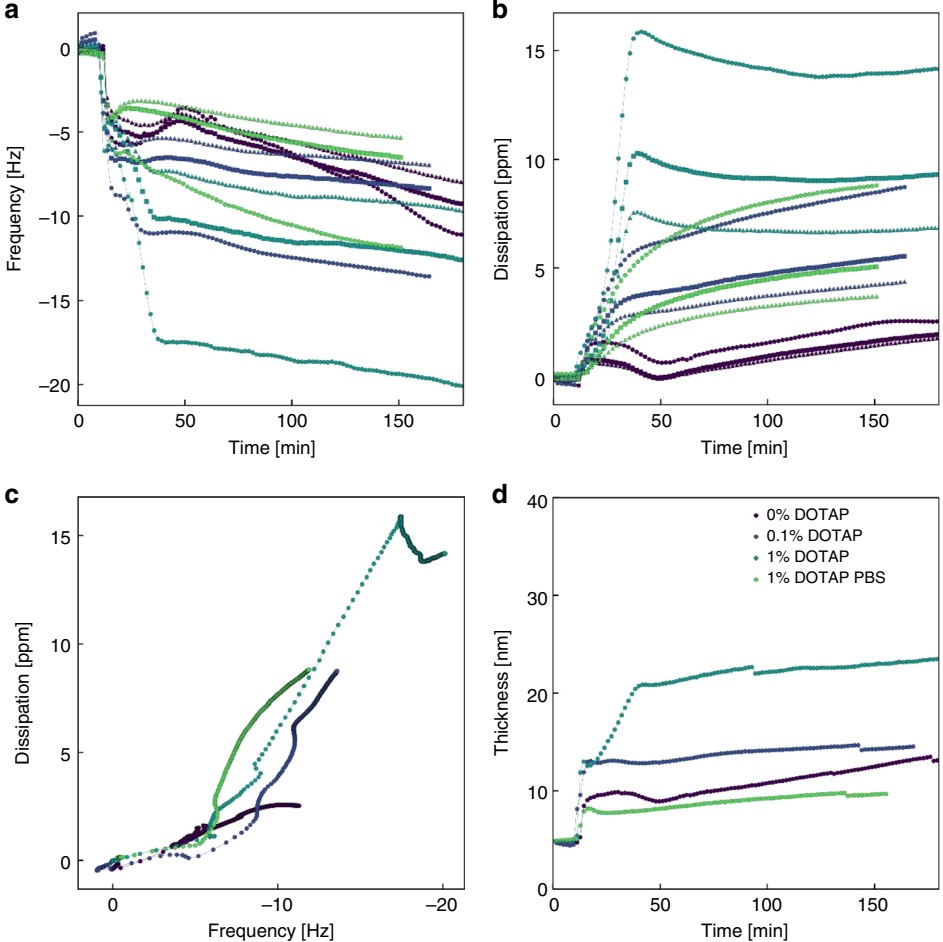

**Fig. 6** Surface measurements during cubosome uptake measured by QCM-D. **a** Shift in frequency normalised by overtone (ΔF/n) with time. **b** Shift in dissipation with time. The interaction of 40% DOPS bilayers with 0%, 0.1% DOTAP, 1% DOTAP in water and PBS are shown in purple, blue, light blue and green, respectively. The overtones $n = 3$, 5 and 7 are indicated by the circle, square and triangle symbols, respectively. **c** Plot of dissipation versus frequency for the 3rd overtone, $n = 3$. **d** Adhered film thickness during cubosome adsorption estimated by Voigt model, using data from 3rd, 5th and 7th overtones

Upon reaching saturation, the concentration gradient driving lipid mixing is nullified, which will negate the proposed model.

The prominent challenge with the use of all nanoparticles in vivo is overcoming the formation of a protein corona upon entering the blood stream. The protein corona greatly alters the effective surface chemistry and topology of the nanoparticle and has been demonstrated to alter their uptake behaviour.[52] Adding to this complexity is the variation in corona composition which has been shown to change with both particle and serum (surrounding) composition.[53] As a consequence, the behaviour of protein coronas is still under investigation and for cubosomes is very much in its infancy. For analogous lipid particles, lipid composition has resulted in varying protein corona outcomes.[54,55] Owing to the prevalence of Albumin in blood serum, and its previous use in in vivo liposome trials in mice,[56] bovine serum albumin (BSA) was utilised to screen the influence of a protein corona on cubosome fusion dynamics. The particle size of MO and MO-DOTAP cubosomes was analysed by DLS in the presence of BSA, shown in Fig. 7a, b. Immediately after introduction to the protein solution in PBS, there was a noticeable increase in particle size and the PDI for both formulations consistent with the formation of a protein corona. A small

additional peak at ~10 nm is believed to correspond to the protein itself. Over a period of several hours, the particle size returned to near the original measurement potentially consistent with protein conformation changes.[57] A shift in zeta potential was also observed in pure water from ~0 mV to – 30 mV for MO cubosomes, and ~30 mV to – 30 mV for 1% DOTAP cubosomes again indicative of the formation of a protein corona.

To examine the influence of the protein corona on cubosome–bilayer interactions, BSA solution was introduced to the microfluidic chamber in between bilayer formation and the introduction of cubosomes to mimic the flow of cubosomes in the blood stream. In all other respects, the bilayer formation followed the same procedure. The protein fouled cubosomes were subsequently introduced and analysed in the same manner as above. The protein:lipid ratio was extremely high at ~70:1 by mass to provide a physiologically relevant protein concentration. For the interaction of MO cubosomes with a DOPC bilayer (i.e., in the absence of charged lipids), the interaction was heavily reduced with very few cubosomes observed to fuse with the bilayer. That said, a representative example of fusion is provided in Fig. 7c. The initial cubosome is observed as the small dim feature at 0 s. Like the previous results, a significant increase in

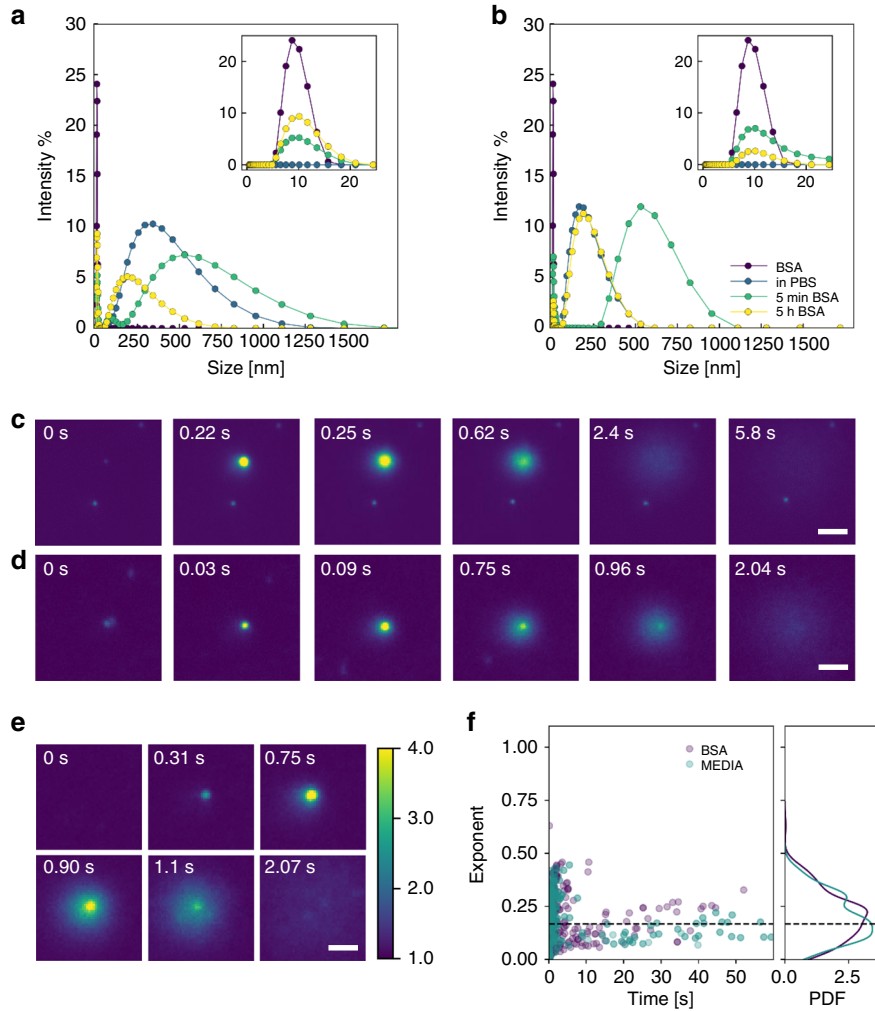

**Fig. 7** Protein corona influence on cubosome–bilayer interactions. **a** DLS measurements on MO cubosomes in BSA. **b** DLS measurements on 1% DOTAP cubosomes in BSA. The results for pure BSA in PBS, cubosome in PBS, cubosome in BSA–PBS for 5 min and for 5 h are shown as the purple, blue green and yellow circles, respectively. **c** TIRF snapshots of a single MO cubosome with DOPC in BSA–PBS environment. **d** TIRF snapshots of a single cubosome fusion event between 1% DOTAP and 40% DOPS 30% CHOL within BSA–PBS environment. Scale bar 1.5 μm. **e** TIRF snapshots of a single cubosome fusion event between 1% DOTAP and 40% DOPS 30% CHOL within complete media environment. **f** Distribution of fusion times and fitted exponents for 1% DOTAP cubosomes with 40% DOPS 30% CHOL bilayer in BSA–PBS (Purple) and complete media (Teal)

fluorescence occurred at ~0.22 s before the fluorescence faded over ~5 s. For the same sample, two stationary cubosomes which did not fuse with the DOPC bilayer can be observed in Fig. 7c as small spots of fluorescence below and above the fusion site. These cubosomes were presumably hindered in their interaction due to the presence of the formed BSA corona.

Remarkably, the cationic DOTAP cubosomes still demonstrated fusion behaviour in line with the results observed in the PBS buffer environment. A representative time-lapse of the fusion is shown in Fig. 7d, showing the attachment over ~30 ms before the fluorescence sharply increased and faded into the SLB over ~2 s. Similar dynamics were observed for the majority of the 1% DOTAP cubosomes, despite the influence of the protein corona. For these samples, we (infrequently) observed interaction involving partial fusion of the cubosome, subsequent recovery and undocking, followed by complete fusion (Supplementary Fig. 13). Initially, the cubosome was observed moving from the top right to the centre left of the frame. Between frames at ~9.76 s and 9.86 s, a small fusion event was observed before the cubosome appeared to regain mobility and

remained stable for a period of ~17 s. Later, at frames ~28.27–31 s, the cubosome fused fully with the bilayer in typical fashion. The double event is perhaps due to some aggregation in the protein environment. The contrast in the rate of redistribution of lipids also draws a parallel to the partial and full lipid mixing observed in virus mediated fusion with red blood cells.[58]

Protein fouling was also investigated in 10% foetal bovine serum (FBS) cell culture media consisting of a mixture of proteins, again revealing the same fusion behaviour. A representative fusion time-lapse is shown in Fig. 7e. The fusion behaviour for the 1% DOTAP cubosomes was examined against the scaling law determined above and is provided in Fig. 7f. A similar analysis was not possible for the fusion of MO cubosomes in BSA–PBS due to the low number of fusion events. The scaling followed the trends observed above, with average exponent very near to the theoretical $n = 1/6$. These results indicate that for 1% DOTAP cubosomes, fusion with the bilayer could still significantly occur in the presence of a protein corona. We note, however, that due to the complexity of protein corona formation,

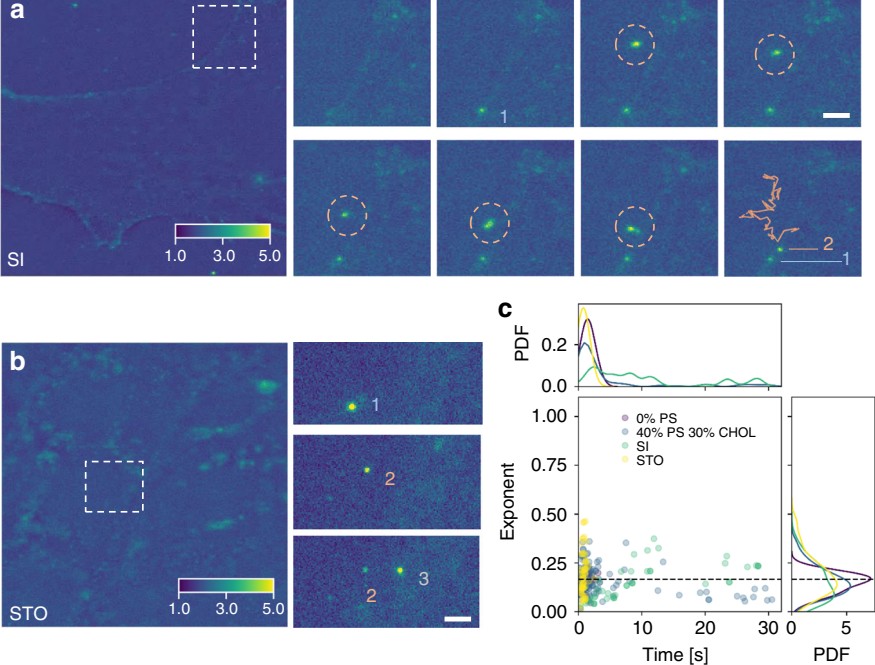

**Fig. 8** Individual cubosome uptake by cells. **a** TIRF image and magnified snapshots for cubosome uptake with small intestine (SI) cells. The times for the snapshots are 0 s, 0.03 s, 0.39 s, 0.45 s, 1.5 s, 1.8 s, 1.86 s, 1.9 s, respectively. The motion and trace of the second cubosome labelled two is highlighted by the pink circles and line, respectively. Scale bar = 1.5 μm. **b** TIRF snapshots for cubosome uptake with STO fibroblast cells. Scale bar = 1.5 μm. **c** Distribution of fusion times and fitted exponents for 1% DOTAP cubosomes with DOPC bilayer (purple), 40% DOPS 30% CHOL bilayer (blue), SI (green) and STO (yellow). The plot includes 126 and 117 unique cubosome interactions for SI and STO, respectively

these interactions would need to be probed under a range of in vivo conditions.[59]

**Fusion dynamics of individual cubosomes with cells**. Clearly, the SLBs provide a simplified model compared to real systems. To initially assess these results against biological cells, analogous experiments were performed with lines of small intestine epithelial cells and STO fibroblasts. These experiments were performed in complete media (consisting of 10% FBS). These samples brought experimental challenges in terms of imaging the cell surface, which is extended above the glass substrate. Nonetheless, individual cubosomes could be observed diffusing through the solution, landing and fading within the cell membrane. Representative snapshots are shown in Fig. 8a, b for the SI and STO cells, respectively. Within Fig. 8a, the attachment of the first cubosome is shown by the first two frames. Then the motion of a second, freely diffusing cubosome is highlighted by the pink circle. In between frame steps, the motion of the cubosome was sporadic before suddenly landing at the cell edge. Once landing, the location of the cubosome was fixed, while the intensity peaks before diffusing towards the background level. During the same period, the gradual fading of the first cubosome was observed. For STO cells, the free tracking of cubosomes above the substrate could not be resolved as clearly; however, the cubosome adsorption and fusion could still be resolved. Examples are shown within the snapshots of Fig. 8b, where a series of three cubosomes landed in succession on the cell surface. Based on the results for SLBs, it is expected that the regions for attachment on the cell surface are likely domains rich in negatively charged moieties. For SI cells, there was a prevalence for attachment at the cell edge which may be rich in substrate-binding proteins or perhaps where the lipid composition is rich in fusogenic lipids. The same fitting routine for fluorescence intensity was performed for these

fusion events, and is plotted within Fig. 8c. The results for the cellular interactions are overlaid against some SLB results. Quite remarkably, there was considerable overlap between the cell and model systems. In both cases, there was a reasonable agreement with both the time for fusion and the fitted exponent for the fluorescence decay. The latter may suggest that the cubosomes are interacting via membrane fusion. It was found that fusion into the STO occurred more rapidly usually within ~ 3 s, compared with SI where some events neared 30 s. In addition to longer fusion durations, the exponent fitting result shows increased heterogeneity. These results may reflect the contrasting lipid compositions of the cell plasma membrane. In comparison to the SLB, the determined fusion times and exponents would reflect a higher cholesterol composition in SI cells, a trait which has been reported in literature.[60] In future work, this protocol will enable the probing of modified and increasingly complex SLBs. This process may also selectively screen cubosome formulations with individual proteins relevant to corona formation towards the development of improved NCs.

## Discussion

The interaction between cubosome NCs and a SLB was observed and characterised by complementary TIRF, liquid AFM and QCM-D experiments, under flow conditions. These experiments allowed the dynamics of fusion to be resolved for individual cubosomes. Instead of acting as a medium for point source delivery, the cubosomes exhibited complete fusion with supported lipid bilayers. The fusion between the two lipid species was greatly accelerated through the inclusion of surface charges. Remarkable contrast was observed in the fusion behaviour of cubosomes with varying lipid domains. When introduced to buffer environments, the cubosome adhered selectively to DOPS rich domains until the membrane composition was sufficiently

MO rich. These results demonstrate preferential attachment using simple lipid variations. The decrease in fluorescence intensity for all samples demonstrated strong agreement to the proposed lipid diffusion model, with an exponent $n \sim 1/6$, which suggests that the cubosome size follows the scaling law $R \sim t^{-1/6}$ during the fusion. This model is expected to hold for non-saturated membranes. It was determined that the formation of a protein corona significantly impacts the capacity for fusion between SLB and cubosomes. Here, cationic cubosomes were found to be more resilient in maintaining the fusion mechanism in BSA and FBS environments. The results for fusion of cubosomes to model supported lipid bilayers correlated strongly with those observed in SI and STO cell lines. This confirms that the interaction of cubosomes with SI and STO cells is via fusion of the cubosome lipid bilayer and the plasma membrane.

## Methods

**Materials**. Monoolein (MO) (97%, Sigma), Pluronic F127 (Sigma), 1,2-dioleoyl-sn-glycero-3-phosphocholine (DOPC) (99%, Avanti), 1,2-dioleoyl-sn-glycero-3-phosphoethanolamine (DOPE) (99%, Avanti), 1,2-dioleoyl-sn-glycero-3-phospho-(1'-rac-glycerol) (DOPG) (99%, Avanti), 1,2-dioleoyl-3-trimethylammonium-propane (DOTAP) (99%, Avanti), 1,2-dioleoyl-sn-glycero-3-phosphoethanolamine-N-(lissamine rhodamine B sulfonyl) (99%, Avanti), octadecyl rhodamine chloride (R18) (Thermofisher), ethanol (AR, Univar) and isopropanol (AR, Univar) were used as received without further purification. Cover glass ($26 \times 60$ mm, No. 1.5H, Marienfeld) were used as substrates. The glass substrates were cleaned by piranha prior to use. Briefly, a 3:1 solution of $H_2SO_4$ (97%): $H_2O_2$ (30%) to which the glass substrates were submerged for 30 min at 85 °C. Following which the glass substrates were successively washed by immersion in pure water. The substrates were stored in pure water until use.

**Supported lipid bilayer production**. Supported lipid bilayer was produced on the hydrophilic glass in a home-built microfluidic channel by solvent exchange process. The microfluidic channel was first filled with a lipid-rich solution i.e., 1 mg mL$^{-1}$ DOPC in ethanol, and then displaced by water at a controlled flow rate. Flow rate is controlled by a syringe pump (NE-1000, PumpSystems Inc.) in the experiments. The molecular images shown in Fig. 1 were prepared using the Charmm-GUI and OpenMM.

**Nanoparticle/carrier preparation**. A typical nanoparticle preparation procedure was as follows: MO (50 mg) was mixed with fluorescent dye R18 (0.25 mg) in ethanol (1 ml). The solution was subsequently dried in a vacuum oven for at least 12 h. The homogeneous lipid mixture was hydrated by a solution of F127 (1 mL, 5 mg mL$^{-1}$). The mixture was dispersed by probe sonication (Branson ultrasonifier 250, 50% Duty Cycle, 5 min), resulting in a 5 wt% dispersion of cubosomes. Variations in cubosome composition, such as DOTAP were introduced in the lipid mixing stage and based on the MO mol %.

**Cell culture**. STO Fibroblast stem cells (ATCC® CRL-1503™) derived from Mus musculus embryo and epithelial cells derived from female homo sapien small intestine (ATCC® CCL-241™) were used as the cell lines. The cell lines were taken from liquid nitrogen. Complete media was used for all cellular work, consisting of DMEM eagle media with 10% FBS and 1% antibiotics (PS). Cells were stored in cryo-conditions using the complete media with 10% DMSO added to avoid water crystals forming. To split and separate cells were spun at 201–290 g at 4 °C to form a cell pellet. The supernatant media was removed before the cells were resuspended in fresh media.

**Synchrotron small-angle X-ray scattering measurements**. SAXS data were obtained using the SAXS/WAXS beamline at the Australian Synchrotron. A typical SAXS experiment used a beam of wavelength $\lambda = 1.033$ Å (12.000 keV) with dimensions 250 μm × 120 μm and a flux of $5 \times 10^{12}$ photons s$^{-1}$. The distance from sample to detector (Pilatus 1-M) was 1.6 m, with a corresponding $q$ range of 0.01 $-0.5$ Å$^{-1}$. A silver behenate standard ($d = 58.38$ Å) was used for calibration. Cubosome solutions were transferred into 96 well plates and sealed, the plates were then mounted directly in the path of the beam, the plate coordinates entered, and an automated scan of each well was performed using an exposure time of 1 s. Upon exposure to the intense X-ray, distinct diffraction rings were observed which were analysed to distinguish the phase and spacing information for the lipid. The data were analysed using the PYFAI package for python.

**Dynamic light scattering (DLS)**. The nanoparticle size and zeta potential were measured using a Malvern Zetasizer Nano (Malvern, UK). The as-produced

cubosome solutions were diluted by a factor of 20 by Milli-Q water. The solutions were placed into polystyrene zeta potential cells (Malvern). The refractive indices for MO and water were 1.46 and 1.33, respectively.

**Microfluidic conditions**. A home-built microfluidic chamber with dimension of (H: 350 μm, W: 15 mm, L: 50 mm) was utilised to form single lipid bilayer on the hydrophilic glass substrate, as sketched in Fig.1a. For the formation of SLBs and the introduction of cubosome NCs, the flow rate was fixed to 50 μL min$^{-1}$.

**Total internal reflection fluorescence microscopy (TIRF)**. TIRF was performed on a Nikon N-Storm super resolution confocal microscope (TIRF 100 ×, 1.49 NA objective lens). The fluid cell was mounted and illuminated by a 488, 561 or 768 nm continuous wave (CW) laser. Within the NIS-Elements AR software, the TIRF mirror position was adjusted until achieving TIR, determined by the appearance of interference patterns of the substrate. The region of interest was collected by an Andor iXon DU-897 EMCCD camera, with a pixel calibration of 0.16 μm per pixel. Image analysis was performed using home-built python codes utilising the open source PIMS, scikit, and TrackPy packages for Python.

**Atomic force microscopy (AFM)**. Liquid AFM was performed on an Asylum MFP-3D or Bruker Dimension Icon. In the former, the as provided fluid-cell light was utilised with an 18 mm glass backing to allow for inverted microscopy. The cantilever (DNP-10) was calibrated against a bare glass substrate before sample imaging. The samples were imaged in AC mode, maintaining a maximal setpoint to reduce force during imaging. In the latter, a home-built fluid chamber was constructed atop of a silicon wafer. The cantilever (ScanAsyst-Fluid+) was calibrated against a bare glass substrate in water phase before sample imaging. The samples were imaged in peak force mode with quantitative nanomechanical mapping (PF-QNM). In both cases, the supported lipid bilayers were formed by mimicking the solvent exchange process manually with micropipettes.

**Quartz crystal microbalance with dissipation (QCM-D)**. QCM-D measurements were performed with a Q-Sense Explorer utilising 50 nm SiO$_2$ sensor disks (QSX-303). The sensor was thoroughly cleaned with isopropanol, water prior to use. The solvent exchange and cubosome introduction was controlled by a peristaltic pump with a flow rate of 50 μL min$^{-1}$. The change in frequency and dissipation was recorded at the 3rd, 5th, 7th and 11th overtones with a sampling interval of 2 Hz.

## Data availability

The data that support the findings of this study are available from the corresponding author upon reasonable request.

## Code availability

The codes used in this study are available from the corresponding author upon reasonable request. The code primarily uses the existing TrackPy package for Python which is available on GitHub.

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

## Acknowledgements

The authors acknowledge the RMIT MicroNano Research Facility for providing access to equipment and resources. C.E.C. acknowledges support from an ARC DECRA Fellowship (DE160101281) and RMIT for a Vice Chancellor's Senior Research Fellowship.

## Author contributions

B.P.D., H.Y. and C.E.C. conceived and designed the research. B.P.D. and H.Y. performed the model experiments. B.P.D. and J.S. performed cellular experiments. B.P.D. and H.Y. analysed the results. B.P.D., H.Y., C.E.C. and C. J. D. wrote the paper and interpreted the results.

## Competing interests

The authors declare no competing interests.
