## [Peer Review File · Nature Communications]

Reviewers' comments:

Reviewer #1 (Remarks to the Author):

This manuscript is a solid study on the interaction of cubosomes with model cell membranes under flow conditions *in vitro*. Cubosomes have been mentioned as possible *in vivo* drug delivery vehicles. However, this has yet to materialize in successful *in vivo* examples. Therefore the detailed study on the physical properties and the behavior of these cubosomes should be studied.

In this work the fusion of cubosomes is proposed with model membranes as a possible pathway for cell uptake. However the presented data is not fully conclusive and the observations could also be explained by docking of cubosomes rather than full fusion. Previously lipid exchange has been shown for cubosomes with membranes and this might also occur here once cubosomes are docked on the membrane.

Another point that would strengthen this manuscript is to perform these experiments but in the presence of blood components. Upon intravenous injection of any nanoparticle a protein corona is immediately formed therefore influencing its fate. So how do these cubosomes interact with blood proteins and how does it influence cell membrane interaction and uptake?

In this study electrostatics are used to induce interaction between the membrane and cubosomes. However a protein corona will influence this, so how does the presented data represent an *in vivo* environment?

on p8 it is said that DOPS regions contain oriented serine residues, however this claim is not backed up with data of literature reference(s)

Fig 3: showing fusion of individual cubosome with a membrane: it has been shown for liposome-SLB fusion that the initial there's an increase of fluorescence due to docking followed by spreading of the signal into the surrounding area due to fusion and eventually the signal fades away. However here the (part of) the signal remains, strongly suggesting docking and (partial) lipid exchange rather than full fusion.

This study would increase its impact if a drug was delivered with a (semi)quantitative bio-readout in cell culture assay, but preferably *in vivo*.

Please add to exp. section on p17 how the hydrophilic glass substrate is prepared

Some references (e.g. 10; 23; 40; 14) are either incomplete or not available to the public yet

In summary: while this study is of interest from a biophysical perspective to gain a better understanding of cubosome model membrane interactions, the chosen approach does not closely enough resemble *in vivo* conditions. Therefore this manuscript, while of interest, should be rejected and published elsewhere

Reviewer #2 (Remarks to the Author):

This is an interesting and important paper addressing the interaction of lipid-based nanoparticles with model as well as living cell membrane. The experiments are very carefully carried out with adequate modelling. I have however some concerns regarding the experimental results and interpretation that I urge the authors to address in order to increase the impact of the paper.

1. Page 5, line 45: Do you believe the cubosomes are intact? The fluorescence merely shows the distribution of the dye not the lipid material! This should be pointed out! Are there any verification experiments that are or should be carried out! Were there any experiments with other dyes?
2. Page 8, line 19: What is the rationale of introducing DOTAP, besides making the cubosome cationic? Why 1% DOTAP?
3. Page 8, line 22: How quantitative is this in terms of number of cubosomes? Are you sure that one light spot is one cubosome?
4. Page 8, line 30: How quantitative is the TIRF data in terms of spatial resolution?
5. Page 11, line 30: Cholesterol can also be regarded as increasing packing!
6. Page 12, line 4: I fully realize the challenges of doing AFM of this system, but I'm not sure if the images add anything. They should be in supporting information.
7. Page 12, line 14: Regarding QCM-D, it would be great to see some attempt to modeling say with the Voigt model. The Figure 5 legend does not specify from which overtone the data is taken. I suggest to plot change in dissipation versus frequency shift as this highlights conformational changes in the system.
8. Page 14, line 18: You say "...attachment at the cell edge which may be rich in substrate binding proteins." Can it not also be lipid domains that favors cubosome attachment?

Response to reviewers - NCOMMS-19-10155-T Fusion dynamics of cubosome nanocarriers with model cell membranes

Reviewer #1 (Remarks to the Author):

This manuscript is a solid study on the interaction of cubosomes with model cell membranes under flow conditions in vitro. Cubosomes have been mentioned as possible in vivo drug delivery vehicles. However, this has yet to materialize in successful in vivo examples. Therefore the detailed study on the physical properties and the behavior of these cubosomes should be studied. In this work the fusion of cubosomes is proposed with model membranes as a possible pathway for cell uptake. However the presented data is not fully conclusive and the observations could also be explained by docking of cubosomes rather than full fusion. Previously lipid exchange has been shown for cubosomes with membranes and this might also occur here once cubosomes are docked on the membrane.

We are grateful for the reviewer's positive feedback on our work and for their suggestions on improving the current manuscript. Below we will address some concerns consecutively.

Fig 3: showing fusion of individual cubosome with a membrane: it has been shown for liposome-SLB fusion that the initial there's an increase of fluorescence due to docking followed by spreading of the signal into the surrounding area due to fusion and eventually the signal fades away. However here the (part of) the signal remains, strongly suggesting docking and (partial) lipid exchange rather than full fusion.

We agree that liposome-SLB fusion has shown spreading and fading of fluorescence due to fusion. The submitted version of Figure 3a highlighted the process over a time frame that did not cover the full fusion process. In fact, upon cubosomes interacting with the SLB we do see the cubosome fluorescence completely dissipate into the SLB. To avoid this confusion, we have updated Figure 3a to include snapshots of the entire fusion process and included time markers in the figure caption. Further, the scaling we observe from equations 1-6 suggests that the cubosomes were not behaving as point release sources for the dye.

That said, on page 13, we also comment that in the latter stages of cubosome interaction we do see a shift towards docking and long-term stability of cubosome features on the substrate. During this stage we also observed partial lipid exchange between the cubosome and the supported bilayer, as shown in the supporting information figure S12.

Another point that would strengthen this manuscript is to perform these experiments but in the presence of blood components. Upon intravenous injection of any

nanoparticle a protein corona is immediately formed therefore influencing its fate. So how do these cubosomes interaction with blood proteins and how does it influence cell membrane interaction and uptake?

The reviewer has raised a valid point regarding the protein corona formed around nanoparticles during injection. The protein corona has plagued application of nanoparticles and greatly complicates the engineering of targeted therapeutics. This issue is still under considerable investigation, and for cubosomes, very little work to date has been performed investigating this issue.

We have now included additional results to help address this concern.

The experiments shown in Figure 7 primarily focus on utilizing bovine serum albumin. Albumin was selected due to its relatively high concentration in the blood. From dynamic light scattering (DLS) experiments carried out in pure water, we can observe fouling of the MO-DOTAP cubosomes by BSA which shifts the zeta potential from $\sim +30$ to ~ -30 mV, consistent with adsorption of the negatively charged protein to the positively charged nanoparticle. Similar experiments on the zeta-potential in a buffer environment were not possible due to the high ionic strength. However, in buffer an increase in particle size was observed again consistent with protein adsorption.

We have subsequently investigated the interaction between the protein fouled cubosomes and model bilayers. We indeed observed a reduction in interaction. The interaction between MO cubosomes and an uncharged DOPC bilayer was the most affected; in general, only $\sim 5\%$ of the interacting cubosomes fused with the bilayer. However, the charged pairs of DOPS-DOPC bilayers and DOTAP-MO cubosomes still resulted in significant fusion behaviour, of a similar magnitude to those observed in PBS environments.

To more closely approximate the environment in the blood we repeated the experiment utilizing complete media inclusive 10% fetal bovine serum, which revealed similar behaviour. Namely, DOTAP-MO cubosomes still resulted in significant fusion events and followed the same scaling as in the absence of protein. These results are also included in Figure 7. Experiments performed utilizing mammalian cells were already conducted in a complete media containing 10% fetal bovine serum where electrostatic interactions should be similarly screened. We have emphasized this in the revised manuscript. Ultimately, we observed a similar scaling in the in-vitro experiments to the model experiments.

These results feature within a new figure - Figure 7 and are discussed in detail on pages 14-16. These new results arguably point to the mechanism behind difficulties in vivo examples raised by the reviewer.

In this study electrostatics are used to induce interaction between the membrane and cubosomes. However a protein corona will influence this, so how does the presented data represent an in vivo environment?

In the original submitted manuscript we had utilized PBS as the model environment. This resulted in a significant reduction interaction due to the electrostatic screening, as suggested by the reviewer. The influence of this is seen in Figure 2-5.

We have now added further experiments to explore the influence of a protein corona, as described just above, primarily seen in Figure 7.

on p8 it is said that DOPS regions contain oriented serine residues, however this claim is not backed up with data of literature reference(s)

For mixed bilayers in buffered environments, selective uptake of the cubosomes was initially only observed in the darker DOPS rich domains until the layer became more saturated by MO. This has been attributed to the enhanced electrostatic interactions between the cationic cubosomes and the anionic DOPS headgroup in these regions. We have speculated that this may be the result of orientated DOPS regions, which is most prominent for TiO₂ substrates but has been shown for mica surfaces (Ref 47,48). It is not clear yet if bilayers formed by solvent exchange would behave exactly as vesicle ruptures studied in the existing literature. Further, we note our glass surfaces are piranha cleaned rather than ozone cleaned which could arguably be involved in the contrast between references. We have added text on page 8, line 11, to clarify this position “In this case, DOPS rich regions possibly contain solution orientated serine residues, inducing the selective attachment observed.” Nonetheless we contend that the mechanism of these selective uptake is secondary to the primary focus of the manuscript.

This study would increase its impact if a drug was delivered with a (semi)quantitative bio-readout in cell culture assay, but preferably in vivo.

The focus of this manuscript is the mechanism of interaction of cubosomes with SLBs, and the relevance of this to their interaction with mammalian cell lines. The use of cubosomes as a drug delivery vehicle is well established and numerous studies have been published demonstrating delivery of cubosome encapsulated drugs to cells. In vivo delivery using cubosomes has also been demonstrated. A list of references is available in Table 2 of reference 14 - *Zhai, J., Fong, C., Tran, N. & Drummond, C. J. . Non-lamellar lyotropic liquid crystalline lipid nanomaterials for the next generation of nanomedicine: design principles, in vivo behaviour, patent landscape, and perspectives. ACS Nano (2019) 13, 6178–6206.* Our view is that this mechanism of interaction is currently one of the most critical missing elements to the use of cubosome delivery agents.

Please add to exp. section on p17 how the hydrophilic glass substrate is prepared

A description of the glass cleaning protocol is now included in the experimental section on page 19.

Some references (e.g. 10; 23; 40; 14) are either incomplete or not available to the public yet

These references should now all be available to the public.

Reviewer #2 (Remarks to the Author):

This is an interesting and important paper addressing the interaction of lipid-based nanoparticles with model as well as living cell membrane. The experiments are very carefully carried out with adequate modelling. I have however some concerns regarding the experimental results and interpretation that I urge the authors to address in order to increase the impact of the paper.

We are grateful for the reviewers' positive feedback on our work and for their suggestions on improving the current manuscript.

1. Page 5, line 45: Do you believe the cubosomes are intact? The fluorescence merely shows the distribution of the dye not the lipid material! This should be pointed out! Are there any verification experiments that are or should be carried out! Were there any experiments with other dyes?

Prior to interaction with the bilayer, we believe the cubosomes are intact. The cubosomes have shown long term stability in solution by DLS and SAXS and should be stable under the flow conditions used. We can further confirm this by the particle tracking results shown in Figure 3f which indicate expected particle sizes.

Upon reaching the SLB we suggest the cubosome gradually disintegrates/fuses with the SLB. We agree the fluorescence is the distribution of the dye. We have altered the text on page 10, line 29 to make this explicitly clear. As part of our modelling we have assumed that the dye is homogeneously distributed throughout the cubosome, which is reasonable for a small dye molecule. Further we emphasize that the scaling observed during fusion corresponds to the model we have used to specifically describe lipid fusion, as opposed to a point source release of dye. Additional dyes have been utilized. We have included example fusion time lapses, intensity curves and scaling for a phospholipid-based dye (18:1) in the supporting information, Figures S9-S11. We note that the phospholipid-based dye showed the same behaviour and scaling further confirming that lipid fusion is being observed. A line of text has been added to feature this on page 10, line 8. "A time-lapse of a single cubosome fusion

event into a DOPC bilayer is demonstrated in Figure 3A. A series of individual fusion events under different conditions is included with Figure S7-8. Results for additional phospholipid-based dyes which show the same behaviour are included in the Supporting Information, Figure S9-11.”

2. Page 8, line 19: What is the rationale of introducing DOTAP, besides making the cubosome cationic? Why 1% DOTAP?

DOTAP was primarily introduced to provide cationic cubosomes having a positive zeta-potential as positively charged particles have previously been shown to have increased interaction with typically anionic cell membranes. It has previously been shown 1% DOTAP significantly increases the positive zeta-potential of MO cubosomes without disrupting the underlying cubic architecture. Ref: *Tran, Nhiem, et al. "Non-lamellar lyotropic liquid crystalline nanoparticles enhance the antibacterial effects of rifampicin against Staphylococcus aureus." Journal of Colloid and Interface Science (2018) 519, 107-118.*

3. Page 8, line 22: How quantitative is this in terms of number of cubosomes? Are you sure that one light spot is one cubosome?

Figure 2f captures the total fluorescence of the imaged area with time. For each bilayer-cubosome pair we see a contrasting rate of increase in fluorescence. We attribute the difference between the intensity profiles being proportional to the number of cubosomes interacting with time, on line 25. In general, we are confident that the cubosomes are not aggregated, from Figure 3f, we observe the bright spots are tracking in accordance to their expected particle size.

4. Page 8, line 30: How quantitative is the TIRF data in terms of spatial resolution?

The TIRF is calibrated at a spatial resolution of 0.158 μm per pixel.

5. Page 11, line 30: Cholesterol can also be regarded as increasing packing!

We have currently attributed the affect to an increase in packing due to cholesterol, hindering the fusion process. We have adjusted the language here to clarify this point.

6. Page 12, line 4: I fully realize the challenges of doing AFM of this system, but I'm not sure if the images add anything. They should be in supporting information.

We have currently decided to keep our AFM image. As they do show the physical interaction through height and modulus not observed in TIRF. We believe this adds to the fusion vs docking argument raised by another reviewer.

7. Page 12, line 14: Regarding QCM-D, it would be great to see some attempt to

modeling say with the Voigt model. The Figure 5 legend does not specify from which overtone the data is taken. I suggest to plot change in dissipation versus frequency shift as this highlights conformational changes in the system.

We are grateful for the reviewer's suggestion and have implemented it in Figure 5. The QCM data was from the 3rd overtone. We have now adjusted the QCM data to include frequency and dissipation shifts from the 3rd, 5th and 7th overtone. In addition, we have included the dissipation vs frequency for $n = 3$. Lastly, we have implemented the Voigt model in attempt to quantify the thickness of the soft film formed during the experiments. The discussion of this figure has been altered to reflect these changes on page 13.

8. Page 14, line 18: You say "...attachment at the cell edge which may be rich in substrate binding proteins." Can it not also be lipid domains that favors cubosome attachment?

The reviewer raises a valid point. We speculated based on the effect of charge we saw in the lipid bilayer experiments that the cubosomes may be greatly influenced by the presence of charge. Our initial discussion pointed towards proteins at the interface, however it may very well be lipid domains as well. We have adjusted our discussion on this page to reflect this.

REVIEWERS' COMMENTS:

Reviewer #2 (Remarks to the Author):

The authors have adequately addressed mine and the other referee's concern. The manuscript has significantly improved and I therefore recommend publication of the manuscript in its revised form.

Reviewer #3 (Remarks to the Author):

After reading this manuscript, the questions raised by Reviewer #1 and the responses of the authors to these questions, I can conclude that the Author's revisions to the manuscript have satisfactorily addressed the concerns of reviewer #1.

NCOMMS-19-10155-T Fusion dynamics of cubosome nanocarriers with model cell membranes

Response to reviewers

Reviewer #2 (Remarks to the Author):

The authors have adequately addressed mine and the other referee's concern. The manuscript has significantly improved and I therefore recommend publication of the manuscript in its revised form.

We are grateful for the reviewer's positive feedback on our work and their previous suggestions to improve our manuscript.

Reviewer #3 (Remarks to the Author):

After reading this manuscript, the questions raised by Reviewer #1 and the responses of the authors to these questions, I can conclude that the Author's revisions to the manuscript have satisfactorily addressed the concerns of reviewer #1.

We are grateful for the reviewer's positive feedback on our work and the previous reviewer's comments on improving our manuscript.

The only comment reviewer #3 had, in comments to the editor, was that a clearer connection between your work and the work of Mittal et al. (2003) Biophys. J. 85:1713-24 should be given due to the similar experimental techniques and similar results.

Considering the reviewers' feedback, we have extended our discussion to describe this connection. The new text is found on page 15, line 41.